# Emotional Impact of COVID-19 and Emotional Eating and the Risk of Alcohol Use Disorder in Peruvian Healthcare Students

**DOI:** 10.3390/nu16172901

**Published:** 2024-08-30

**Authors:** Jhon Alex Zeladita-Huaman, Juan Pablo Aparco, Eduardo Franco-Chalco, Luz Nateros-Porras, Sonia Tejada-Muñoz, Denices Abarca-Fernandez, Iris Jara-Huayta, Roberto Zegarra-Chapoñan

**Affiliations:** 1Academic Department of Nursing, Faculty of Medicine, Universidad Nacional Mayor de San Marcos, Lima 15001, Peru; 2Academic Department of Nutrition, Faculty of Medicine, Universidad Nacional Mayor de San Marcos, Lima 15001, Peru; juanpabloaparcobalboa@gmail.com; 3Psychology Department, Pontifical Catholic University of Peru, Lima 15008, Peru; eduardo.franco@pucp.edu.pe; 4Teaching and Research Office of the Healthcare Integrated Network Directorate of Downtown, Lima 15023, Peru; luzprevencion@gmail.com; 5Institute of Tropical Diseases, Academic Department of Public Health, Universidad Nacional Toribio Rodríguez de Mendoza de Amazonas, Chachapoyas 01001, Peru; sonia.tejada@untrm.edu.pe; 6Faculty of Nursing, Universidad Nacional del Altiplano, Puno 21002, Peru; denicesabarca1@gmail.com; 7Faculty of Health Science, Universidad Nacional de San Cristóbal de Huamanga, Ayacucho 05002, Peru; iris.jara@unsch.edu.pe; 8Faculty of Health Science, Universidad María Auxiliadora, Lima 15408, Peru; rob.zegarra@gmail.com

**Keywords:** depression, mental health, COVID-19, alcohol drinking, health occupations, students

## Abstract

Background: This study aimed to explore the association between the emotional impact of COVID-19 and emotional eating and the risk of alcohol use disorder among Peruvian health science students. Methods: We conducted a cross-sectional analytical study in which an online questionnaire was administered to 456 health science interns from four cities in Peru. We used the COVID-19 Emotional Impact Profile questionnaire, Mindful Eating Questionnaire, and Alcohol Use Disorders Identification Test. Spearman’s correlations were calculated and two multiple linear regression models were developed. Results: 68.4% of the participants were emotional eaters and 8.6% reported low-risk levels of alcohol use disorder. Based on the results of the first model, the overall emotional impact of COVID-19, being overweight or obese, depression and anxiety levels, and living with only one parent were factors associated with emotional eating. The results of the second model showed that the level of depression, living with just one parent, living alone, sex, and number of months as an intern were factors associated with the risk of alcohol use disorder. Conclusions: To reduce emotional eating and the risk of alcohol use disorder among interns, universities should implement interventions aimed at reducing the emotional impact of COVID-19 and provide nutritional counseling.

## 1. Introduction

By October 2023, over 6.8 million people worldwide had died as a result of the COVID-19 pandemic [1], which also caused economic delays and altered many aspects of society, including education and health [2]. Regarding its repercussions on mental health, the World Health Organization emphasized that the prevalence of mental disorders such as depression and anxiety increased by 25% during the first year of this global crisis [3]. Likewise, the long-term effects of COVID-19 affected general mental health and caused cognitive decline, memory impairment, anxiety, sleep abnormalities, and depressive-like behavior [4].

According to studies conducted during the most recent epidemics, university students pursuing health science careers have experienced an increase in mental health problems [5]. For example, a meta-analysis revealed that the prevalence of mental health issues such as depression (41%), anxiety (38%), stress (34%), sleep disorders (52%), and burnout (38%) in this population increased significantly during the COVID-19 pandemic [6]. These problems have also been detected in university students from various regions of Peru [7], particularly those working in the health sector [8]. The impact of COVID-19 pandemic on the mental health of Peruvian university students was determined using the scale for the Emotional Impact Profile of COVID-19 (EIP-COVID-19) [9], and it was especially reported that there were differences in the dimensions of fear and depression according to sex [10]. However, in university students, little research has been found that analyzes the impact of increased psycho-emotional problems because of the COVID-19 pandemic’s effect on lifestyles such as in terms of food consumption [11,12] and adoption of harmful habits, such as alcohol consumption [13,14].

Diet has a significant impact on mental health and is regarded as an environmental factor [15]. However, little is known about the relationship between negative emotions and diet during a pandemic [16]. According to some studies, emotions influence the quantity and quality of food we eat because they control hedonic processes, which regulate energy balance, and homeostatic processes, which regulate food pleasure, reward, and palatability [17,18]. Furthermore, negative emotions affect eating behaviors, weight changes, and underlying psychological processes [18]. During the COVID-19 pandemic, for instance, emotional distress was correlated with emotional eating [19]. In this context, the concept of emotional eating emerged, which is defined as the propensity to overeat to cope with negative emotions such as depression, anxiety, and stress [20]. During the COVID-19 pandemic, 27.4% of university students were emotional or very emotional eaters [21]. In a study conducted in the Kingdom of Saudi Arabia, one in two women was found to be an emotional eater [22]. Another study reported that five out of six Peruvians were emotional eaters [23].

Regarding the studies carried out on university students during the COVID-19 pandemic, there is an inconsistency among the reporting of the association between emotional food consumption and psycho-emotional aspects. While some studies highlight its association with depressive symptoms [24], anxiety [11] and stress [25], other studies reported contradictory results between the association of depression and anxiety with this construct [22,26]. Furthermore, few studies have been found that evaluate its association with academic aspects, nutritional status, and the family environment [12,22].

However, evidence regarding alcohol consumption during the COVID-19 pandemic compared with pre-lockdown use is contradictory. While one study reported an increase [27] another revealed a decrease [28], particularly in low-and middle-income countries [29]. During stressful events, people frequently experience an upsurge in mental health issues (e.g., depression and anxiety) and substance abuse [30]. Moreover, factors such as non-compliance with quarantine restrictions, number of individuals in the household, sex, age [28], emotional distress [29], and depression [31] were found to be associated with alcohol consumption during the COVID-19 pandemic. Meanwhile, studies conducted prior to the COVID-19 pandemic suggested that 25 to 27.5% of Peruvian university students were at risk of alcohol abuse [31,32].

Regarding the factors associated with the consumption of alcoholic beverages in university students during the COVID-19 pandemic, researchers generally agree that the consumption of alcoholic beverages is greater in men than women and that it is associated with age [14,33,34,35]. Regarding psycho-emotional aspects, some studies highlight its association with depression [36,37], anxiety and stress [13,33]. This disagrees with a study that reports that there is no association with these psychological variables [14]. Likewise, there are few studies that analyze its association with the family environment; in this regard, one study reports its association with family dynamics [38] and another study with living alone [35].

Although there are several studies that analyze the association between depression and anxiety with the emotional consumption of food and the risky consumption of alcoholic beverages in university students, these studies were carried out on students from different years of study and from various areas of knowledge [11,12,33]. However, no studies have been found that were carried out on university health sciences students who were in their last year of studies, a period in which they carry out internships (professional internships) in hospital and community settings. Likewise, no studies have been found that consider other variables that characterize the psycho-emotional impact of this recent pandemic, such as fear and anger.

This research will provide information that allows us to understand the relationship between the emotional impact that the COVID-19 pandemic had on the emotional consumption of food and risky alcohol consumption in health sciences interns from different careers who were carrying out their professional practices after having been in quarantine and receiving non-face-to-face education for the previous two years, in a context where they faced not only the fight against the pandemic and but also the restart of pre-professional practices in person to address the social determinants of health. This information is key to joining efforts and taking action to protect this population. Thus, this study aimed to explore the association between the emotional impact of COVID-19 and emotional eating and alcohol consumption among Peruvian healthcare students.

## 2. Materials and Methods

### 2.1. Sample and Procedures

This cross-sectional and analytical study was conducted in four cities in Peru (Lima, Amazonas, Puno, and Ayacucho). Authorization to conduct this study was obtained from the healthcare networks that coordinated academic institutions and healthcare providers. For the regions of Amazonas and Ayacucho, authorization was specifically requested from the universities attended by the interns. Informed consent was obtained online from all participants. Data collection was carried out using an online questionnaire designed with Google Forms (Version 1.0), which was distributed via email and WhatsApp (version 2.23.12.75) to the list of interns provided by the coordinators of each university. Data were collected between February and May 2022.

To ensure that only participants who met the eligibility criteria completed the self-administered questionnaires, the following strategies were implemented: 1. When access to the list of potential participants was obtained, we verified the information with the student through a phone call before sending the questionnaire. 2. During this initial call the importance of their participation in the study was explained, emphasizing the need for honest responses to the questionnaire. We also inquired about the likely date they could complete the questionnaire, and upon completion, a thank you email was sent to them.

### 2.2. Population and Selection Criteria

The study population consisted of senior students older than 18 years pursuing health science careers from both public and private universities and working as interns in healthcare facilities in the selected cities. Students who did not have the time to complete the questionnaire or who were diagnosed with a mental health disorder were excluded from the study. Students with a history of mental illness diagnosed before the pandemic were excluded, considering that a history of mental illness can be a confusing factor affecting both the emotional impact of COVID-19 and emotional eating.

Assuming that the effect of the emotional impact of COVID-19 on emotional eating was minimal (f2 = 0.02) and using a type I error probability of 0.05 and a minimum statistical power of 80%, the sample size was estimated to be 311 individuals. However, to account for sample loss, we only collected data from 450 interns. As the study was conducted in four cities, the sample quota was distributed as follows: 150, 100, 100, and 100 interns from Lima, Puno, Amazonas, and Ayacucho, respectively. Non-probability convenience sampling was employed in this study.

### 2.3. Measurement Tools

For the data collection, we designed an online questionnaire using Google Forms. In the first part of the questionnaire, participants were briefly informed of the purpose of the research and requested to sign an informed consent form. The second part included a series of questions to collect participants’ sociodemographic (age, sex, marital status, occupation, family composition, and relatives with whom they lived), epidemiological (history of COVID-19), and academic data (type of university, career, and number of months as an intern). Participants were also asked about their weight and height at the time of completing the questionnaire.

#### 2.3.1. COVID-19 Emotional Impact Profile

Regarding the assessment of mental health, the Emotional Impact Profile questionnaire was used to determine the emotional impact of the COVID-19 pandemic [9]. This questionnaire was validated among Peruvian university students. It consists of 25 questions that measure individuals’ levels of anger, resentment, fear, anxiety, stress, and depression, as these were the primary mental health issues reported during the recent pandemic. A confirmatory factor analysis was performed on the present sample to assess the construct validity and reliability of the model, and it demonstrated an excellent fit (X^2^/*df* = 0.74, CFI = 1.00, TLI = 1.00, RMSEA = 0.00, SRMR = 0.04) with a Cronbach’s alpha of 0.86.

#### 2.3.2. Mindful Eating Questionnaire (MEQ)

We used the Mindful Eating Questionnaire (MEQ) to evaluate emotional eating, which characterizes a nonjudgmental awareness of the physical and feelings related to eating. This scale was first validated in a sample of obese and overweight Spanish individuals [39]. Later, it was validated among Chilean university students, demonstrating its applicability and adequate psychometric properties [40]. This Likert-type scale includes 10 items with four response options: 0 = never/rarely, 1 = sometimes, 2 = often, and 3 = usually/always. The scale inquiries about usual eating behavior in response to negative emotional states without specifying a time frame. These questions aim to determine whether food intake is influenced by various negative emotions. The model’s construct validity and reliability were also assessed using a confirmatory factor analysis (CFA), which showed an excellent fit to the data (X^2^/*df* = 1.12, CFI = 0.99, TLI = 0.99, RMSEA = 0.16, SRMR = 0.05), with a Cronbach’s alpha of 0.91. To establish the categories of emotional eaters, we used the ranges defined in the initial validation study. Participants were classified as follows: non-emotional eaters (less than 6 points), indicating that emotions have little to no influence on their eating behavior; low-emotional eaters (6 to 10 points), indicating those who are not highly emotional regarding their eating habits but feel that certain foods influence their willpower; emotional eaters (11 to 20 points), characterized by a significant influence of emotions on their eating behavior; and very emotional eaters (21 to 30 points), where their emotions consistently revolve around eating, potentially leading to disordered eating behaviors if coping strategies are not implemented [39]. 

#### 2.3.3. Alcohol Use Disorders Identification Test (AUDIT)

In the final part of the questionnaire, participants were asked about their risk of alcohol use disorder using the AUDIT, which was tested by Segen et al. [20] among Spanish-speaking university students. The original version of the AUDIT, developed by the WHO, consists of three factors: alcohol consumption, drinking behavior, and alcohol-related problems. Participants respond to each question by indicating the frequency of alcohol use and/or the experience of symptoms related to problem drinking on a scale of 0 (“never”) to 4 (“4 or more times per week”), which results in a maximum possible score of 40. Higher scores indicate a higher risk of drinking problems. CFA was also performed to determine the model’s construct validity and reliability, which showed an excellent fit to the sample (X^2^/*df* = 0.76, CFI = 1.00, TLI = 1.00, RMSEA = 0.00, SRMR = 0.06), with a Cronbach’s alpha of 0.81.

All the scales used in this study were validated by Spanish speakers. Moreover, in this study, the content validity of the designed instrument was verified by a group of mental health experts. The completion time for the questionnaire was between 13 and 20 min.

### 2.4. Analysis

To achieve this study’s goals, we first conducted a descriptive analysis of the variables. We then computed Spearman’s correlations between continuous variables to identify bivariate associations, because some variables did not show a normal distribution. Furthermore, several Kruskal–Wallis tests and Mann–Whitney tests were conducted to determine the relationship between the categorical variables, levels of emotional eating, and risk of alcohol use disorder. In the Kruskal–Wallis test where statistically significant differences were found, we performed post hoc comparisons using Bonferroni correction.

To achieve a robust and reliable identification of the variables most closely associated with emotional eating and the risk of alcohol use disorder, we employed a comprehensive statistical modeling approach involving two distinct multiple linear regression models tailored to each dependent variable. Both models included age and sex as control variables, given their common association with emotional eating and alcohol use disorders in the literature. Controlling for these factors helped isolate the effects of the primary variables of interest.

We utilized a forward stepwise variable selection algorithm to refine our models and ensure they captured the most explanatory variables. This method systematically added variables to the model based on specific criteria, optimizing the model’s explanatory power. The process began with an empty model containing only the intercept. Variables were added one at a time, with each step evaluating which variable most significantly improved the model’s ability to explain the variance in the dependent variable. The selection of variables was based on the Akaike Information Criterion (AIC) and *p*-values from F-tests, with a variable included in the model if it significantly reduced the AIC, indicating a significant improvement in model fit. This process was repeated iteratively, with each new variable being evaluated and added if it met the inclusion criteria, and the algorithm being terminated when no additional variables significantly improved the model.

To ensure the validity of our regression models, we conducted several diagnostic tests. We assessed the normality of residuals using visual inspections and statistical tests, applying transformations to the dependent variables using Tukey’s ladder of power when necessary. We evaluated the homoscedasticity of residuals to confirm constant variance across all levels of the independent variables. We checked for multicollinearity among the independent variables using variance inflation factors (VIFs) to ensure that no two variables were highly correlated, which could distort the regression coefficients. Additionally, we used Cook’s D tests to identify and address potential outliers and influential data points that could unduly affect the model’s estimates. R software (version 4.2.1) was used to perform all statistical analyses.

## 3. Results

A total of 550 potential participants were invited to participate in this study, 94 of whom did not complete the questionnaire. Our study included 456 participants, with an average age of 26 years. The majority (83%) were female. Most participants were single (89%), with approximately 10% married or cohabiting, and 1% divorced. Regarding living arrangements, 37% lived with both parents, 22% lived with one parent, 17% lived with other family members, and 24% lived alone. In terms of COVID-19 status, 60% had experienced mild cases, 4% had severe cases, and 36% had not been infected at the time of the study. Approximately half of the participants were working while studying, whereas the other half were solely focused on their studies. Among them, 68% attended public universities and 32% attended private universities. The fields of study varied: 56% were in nursing, 15% in human medicine, 14% in medical technology, and the remainder were in biochemistry, pharmacy, nutrition, obstetrics, dentistry, and psychology.

Regarding emotional eating, 31.6% of participants were categorized as non-emotional eaters, 35.3% as low-emotional eaters, 28.5% as emotional eaters, and 4.6% as very-emotional eaters. Table 1 shows the medians of emotional eating and risk of alcohol use disorder for the different sociodemographic variables analyzed. Regarding emotional eating, there were significant differences in terms of sex (*p* = 0.005), marital status (*p* = 0.002), nutritional status (*p* < 0.001), history of COVID-19 (*p* < 0.001), and family composition (*p* = 0.02). More specifically, the median emotional eating was found to be higher in women than in men. Moreover, post hoc comparisons revealed that emotional eating scores were higher for single participants than for married ones (*p* = 0.02). Considering nutritional status, people who were overweight and obese revealed higher scores in emotional eating than participants with normal weight (*p* < 0.001). Regarding history of COVID-19, emotional eating scores were higher for individuals who experienced severe symptoms than for those who had not yet been infected (*p* < 0.001) and those who had mild symptoms (*p* < 0.001). Finally, regarding family composition, emotional eating scores were lower for participants living with other relatives than for those living with just one parent (*p* = 0.02), with this being the only difference observed in this variable. 

In terms of alcohol consumption, 90.4% of participants were categorized as abstainers, 8.6% as low-risk drinkers, and 1.1% as risky drinkers. As shown in Table 1, sex (*p* < 0.001), history of COVID-19 (*p* = 0.002), and family composition (*p* = 0.008) were statistically significant variables. Specifically, the mean AUDIT score was higher in men than in women. Regarding history of COVID-19, AUDIT scores were higher for participants who experienced severe symptoms than for those who had not yet been infected (*p* = 0.03) and those who had mild symptoms (*p* = 0.05). Finally, concerning family composition, alcohol consumption scores were higher for participants living alone than for those living with both parents (*p* = 0.008) and other relatives (*p* = 0.002). Similarly, AUDIT scores were higher for participants living with just one parent than for those living with other relatives (*p* = 0.02).

Table 2 presents some of the descriptive statistics for the variables under analysis as well as the correlation matrix between age, number of months as an intern, emotional eating, risk of alcohol use disorder, and various COVID-19 emotional impact dimensions. Emotional eating was negatively, significantly, and weakly correlated with age and positively, significantly, and weakly correlated with the risk of alcohol use disorder. Similarly, emotional eating was positively, significantly, and moderately to strongly correlated with all COVID-19 emotional impact dimensions, with the strongest correlations being with anxiety, stress, depression, and overall emotional impact. The risk of alcohol use disorder was positively and weakly correlated with anger, anxiety, stress, depression, and overall emotional impact.

The resulting model for predicting emotional eating is shown in Table 3. The COVID-19 emotional impact scale score, anxiety, depression, nutritional status, and family composition were selected, indicating their important contributions to the present model. The standardized coefficients indicated that the variables most strongly associated with emotional food consumption were scores on the COVID-19 emotional impact scale (*B* = 0.24, *p* = 0.002) and depressive symptoms (*B* = 0.24, *p* < 0.001). In addition, the results showed that people who were overweight or obese had higher levels of emotional food consumption than those with normal weight (*B* = 0.14, *p* < 0.001). Similarly, participants with higher anxiety levels reported higher emotional consumption levels (*B* = 0.17, *p* = 0.005). Finally, regarding family composition, participants who reported living with only one parent also reported higher consumption levels than those who reported living with both parents (*B* = 0.10, *p* = 0.012). The model had an R^2^ of 0.43, indicating that the variables included in the model explained 43% of the variability in the emotional consumption of food.

The following linear regression model examined the relationship between risky alcohol use and several variables: age and sex (control variables), depression, length of internment, and family composition (Table 4). The results of the selected model indicated that the significant variables were sex (*B* = 0.33, *p* < 0.001), depression (*B* = 0.21, *p* < 0.001), length of hospitalization (*B* = 0.11, *p* = 0.015), and family composition (living with one parent vs. living with both parents: *B* = 0.11, *p* = 0.018; living alone vs. living with both parents: *B* = 0.12, *p* = 0.016). The standardized coefficients indicated that sex was most strongly associated with risky alcohol use, followed by depression. The results also showed that people living with a single parent or alone and those who reported more depression had higher levels of risky drinking. Regarding the dimensions of anger, university type, and anxiety, although they were included in the model, they did not have a statistically significant effect on risky alcohol consumption. The model had an R^2^ of 0.20, indicating that the variables included in the model explained 20% of the variance in alcohol use risk.

## 4. Discussion

This study investigated the relationship between the emotional impact of COVID-19 and emotional eating and alcohol consumption among healthcare science interns. To this end, we developed two regression models using sex and age as the control variables. In the first model, the overall emotional impact of COVID-19, depression and anxiety levels, nutritional status, and family composition were associated with emotional eating. In the second model, depression levels, family composition, sex, and number of months as an intern were reported as factors associated with the risk of alcohol use disorder.

Although the correlational analysis revealed that the COVID-19 emotional impact scale score and all its dimensions were correlated with emotional eating, the multivariate analysis model only selected the COVID-19 emotional impact scale score depression, and anxiety levels as associated factors. This finding is consistent with the results of a study conducted in Italy, in which the authors found that greater control over overeating during lockdown was associated with not being anxious [41]. Similarly, in another study, depressive symptoms in first-year students at a Mexican university were associated with emotional eating in both men and women [24]. According to another study conducted among nursing students in Turkey, depression, anxiety, and stress had a direct and statistically significant relationship with emotional eating [25]. Likewise, in Spain, a study carried out before the COVID-19 pandemic in first- and second-year nursing students reports that the trait of anxiety was negatively related to emotional food consumption [11]. Furthermore, a study conducted in Egypt on students, employees, and medical staff at a medical school reported that anxiety was a predictor of emotional food consumption [12]. However, a study conducted on health science students in Turkey that performed a structural equation modeling analysis found that anxiety had a significant direct effect on emotional food consumption, but depression did not because it could not remain as another affective component in the model [26].

The finding of this study is that the total score of the emotional impact of the COVID-19 pandemic had a direct effect on the model of emotional food consumption, evidence that the combination of depression, anxiety, stress, and anger that was experienced during the pandemic had an impact on food consumption. Moreover, the correlation between emotional eating, depression, and anxiety suggests that the negative emotions caused by the COVID-19 pandemic led to an increase in eating as a coping mechanism. This may be explained by the fact that emotional eating is the failure to distinguish between hunger and the desire to eat to cope with negative feelings [42]. Additionally, compulsive eaters, when placed in circumstances where they must live up to high standards or fulfill the expectations of others—a situation that students can experience during an internship—experience emotional distress characterized by anxiety and depression [43]. This conclusion is relevant because people who are not in a good mood tend to overindulge in high-calorie foods, especially sweets and chocolates, because they believe that this will make them feel better; however, this could result in being overweight or obese [22,44].

Based on the psychosomatic theory, several studies have reported an association between emotional eating and increased intake of unhealthy foods such as snacks and fast food [45,46]. This is because emotional eaters tend to eat these foods to reduce the intensity of their negative emotions and cope when they do not have other coping mechanisms. Eating palatable foods, typically foods heavy in sugar or fat, produces instant pleasure and rewards (positive emotions) that can reduce the effects of stress [45,46]. Additionally, because eating triggers emotions, the palatability of high-carbohydrate or high-fat foods is key to emotion regulation [47]. Another reason for choosing this type of food to cope with emotions is its availability and accessibility. For example, unhealthy foods are ready–to–eat and easily accessible, whereas fruits and vegetables are less readily available. Moreover, street food provides individuals at all economic levels with various dietary options [48].

Furthermore, being overweight or obese was associated with emotional eating among health science interns. This correlation has also been reported in a study conducted among first-year students at a Mexican university before the COVID-19 pandemic. In this study, the authors used the Emotion- and Stress-Related Eating Questionnaire and found that body mass index (BMI) was associated with emotional eating and that the latter acted as a mediator between depression and BMI, adjusted for age in both sexes [24]. Similarly, a study conducted in the Kingdom of Saudi Arabia during the COVID-19 pandemic that employed the Dutch Restrained Eating Scale reported that university students or graduates with higher BMIs had a higher risk of being emotional eaters [22]. Additionally, according to studies conducted in Turkey, BMI was directly associated with emotional eating scores in adults quarantined during the pandemic [49], and the percentage of emotional eaters was higher among overweight and obese individuals [50]. However, a study carried out in Egypt reports that BMI is not a predictive factor of emotional food consumption [12]. These results demonstrate that obesity is a biological predictor of emotional eating. However, it is crucial to keep in mind that obese individuals may have a history of emotional eating, which may cause them to gain weight over time. 

Similarly, in this study, students living with only one parent obtained higher emotional eating scores than those living with both parents. This finding indicates that interns who did not receive adequate family support sought refuge in food to deal with psych- emotional problems. The percentage of students found to be emotional eaters in this study coincides with that reported in a study in which an online questionnaire was administered to adults in Turkey during the first year of the COVID-19 pandemic [50]. However, this was higher than that reported in another study conducted among female students and graduates in the Kingdom of Saudi Arabia [22]. 

Sex had a moderate to strong impact on alcohol consumption. Men reported higher levels of alcohol consumption than women, which is in line with research carried out in other parts of the world in the general population [28,51] and with respect to university students [33,34,35,52]. We found that students who had been interns for longer reported higher levels of risk for alcohol use disorder, although this effect was low. This may be explained by the fact that young interns turned to alcohol for comfort as the demands of their preprofessional training increased as the internship progressed. However, this is only a short-term solution that can eventually become troublesome or result in dependence. 

In terms of family composition, students living with only one parent or by themselves reported higher levels of risk of alcohol use disorder than those living with both parents. This is consistent with the results of a previous study, in which most participants were from Europe and Latin America. According to these studies, larger household size was associated with a decline in problematic alcohol consumption among young adults during the early stages of the COVID-19 pandemic [28]. Similarly, living alone was reported to be a positive predictor of higher levels of harmful alcohol consumption in university students in Germany [35], while for university students from Mexico they reported that family dynamics are a predictor of alcoholic beverage consumption [38].

Finally, correlational analysis revealed that the COVID-19 emotional impact scale score and its four dimensions correlated with the AUDIT scale. However, according to multivariate analysis, only depression was found to have a low-to-moderate effect in predicting this variable. Similarly, in a study conducted among Slovak university students, depression was found to have a positive effect on alcohol consumption [53]. In addition, alcohol consumption has been associated with depression in Brazil [54] and United States [37]. According to another study conducted in the United States, participants with moderate-to-severe depression were more likely to drink alcohol than those with no or mild depression [55]. Moreover, a study carried out on university students in Serbia reported that a greater frequency of alcoholic beverage consumption was associated with symptoms of anxiety, stress, and depression [36]. On the contrary, a correlational study carried out in Serbian university students reported that anxiety and stress show a correlation with the consumption of alcoholic beverages; however, depression was not correlated [13]. Furthermore, another study carried out on Hungarian university students reported that depression, anxiety, and stress are not associated with the consumption of alcoholic beverages [14]. 

During their internships, health science students had to manage situations that could exacerbate their mental health issues, such as leaving their hometown, working to support their family, and completing their studies amid a pandemic that we had never seen before. Combined with depression, these factors could lead to increased alcohol consumption. However, during the COVID-19 pandemic, academic stress was associated with poor academic results among university students [56]. The exploratory model used allowed us to evaluate the predictive value of the analyzed variables. Therefore, to deepen the relationships found in the present study, other statistical models such as the structural equation model must be used.

Identifying the factors that influence emotional eating in university students, especially health science interns, is key to preventing and treating not only being overweight and obesity but also other related medical issues, especially among young populations. Based on the findings of this study, when developing effective coping mechanisms and providing nutritional education, variables such as anxiety, depression, family support, and university support during internships should be considered. This will contribute to the development of the personal, social, and environmental conditions necessary for students to have optimal psychological and physiological well-being, and thus, a better quality of life.

This study provides important insights into the association between psycho-emotional disturbances during the COVID-19 pandemic and emotional eating, as well as the risk of alcohol consumption (predisposition to adopting harmful habits) among university students in health sciences who were beginning pre-professional practices (internships) in primary healthcare settings after two years of virtual classes. By providing robust statistical evidence of this association, we contribute to the understanding of the long-term impact of mental health on lifestyle in a post-pandemic scenario. Furthermore, in terms of practical implications, highlighting the impact of the psycho-emotional dimension on the quality of life of healthcare students could support the urgent need to implement public policies and interventions aimed at improving the psychological well-being of health sciences students.

Among the interventions that have been reported to significantly improve the psychological well-being of undergraduate students are: the online-based Acceptance and Commitment Therapy developed in Malaysia [57], an intervention conducted in Indonesia that promotes mindful reflection on the positive aspects of life and encouraged participants to “count their blessings”, and programs based on game theory that optimize emotional strategies, such as increasing happiness levels [58]. Likewise, some strategies are identified with social participation and leisure as central elements both individually and collectively [59]. It is recognized the that autonomy and the ability of individuals to act upon themselves are the starting points for supporting the actions that young people undertake in transforming their own lives. Strategies aimed at addressing emotional eating and reducing alcohol consumption in young people must empower them to recognize themselves as socio-historical subjects and agents of change.

Among the strengths of the study we can highlight: (1) having a sample that was made up of health sciences students from different disciplines such as medicine, nursing, psychology, midwifery, medical technology, nutrition, who came from public and private universities of four cities in Peru, which allowed for greater heterogeneity in the results; (2) the confirmatory factor analysis carried out on the emotional impact profile scale of COVID19 allowed us to demonstrate that the instrument has adequate psychometric properties and that the findings are more reliable and statistically robust.

This study had some limitations. First, because the data were collected via a self-report questionnaire, the findings could have been influenced by social desirability bias, and we may not be 100% sure that the questionnaires were completed by the individuals who were invited to participate in the study. The same applies to honesty when answering questions, especially in the AUDIT questionnaire. Second, the non-probability sampling method employed does not guarantee that the participants represented the population under study. Third, because this was a cross-sectional study, the results cannot be generalized and may vary because of the dynamic effects of the pandemic. Therefore, further longitudinal studies are warranted. The students tested were not diagnosed with pre-existing eating disorders, this could have affected the reported results because eating disorders could easily lead to emotional eating. Finally, the scales employed can only be used as screening tools and not as diagnostic tools. 

## 5. Conclusions

In this study, which was conducted on health science students during their internship in four cities in Peru, the overall emotional impact of COVID-19, levels of depression and anxiety, nutritional status, and family composition were associated with emotional eating, whereas the level of depression, family composition, sex, and number of months as an intern were associated with alcohol consumption.

According to our findings, the faculties of health sciences at universities should provide students with psycho-emotional support and nutritional counseling so that they can develop psychological tools for managing their anxiety and depression and build healthy eating habits; it is necessary to consider the different family composition situations in students for support interventions, since the findings show greater vulnerability for students who live with a single parent or alone.

Similarly, higher education institutions should implement effective intervention programs to prevent and manage alcoholic beverage consumption among students. Specifically, the emotional eater instrument is a scale of usual food consumption behavior in the face of negative emotions; it does not quantify the amount of food consumed or inquire about the quality of food; however, it has been applied in various feeding studies during the pandemic because it reflects the eating behavior of the population in the face of a global event such as COVID-19.

## Figures and Tables

**Table 1 nutrients-16-02901-t001:** Comparison of medians of emotional eating and alcohol consumption according to sex, marital status, occupation, history of COVID-19 infection, family composition, and type of university.

Variables	Emotional Eating		Risk of Alcohol Use Disorder	
	Med (IQR)	Est	Med (IQR)	Est
Sex				
Female	8.00 (8.00)	7.72 *	1.00 (3.00)	30.17 ^‡^
Male	6.00 (7.75)		3.00 (6.00)	
Marital status				
Single	8.00 (8.00) ^a,c^	12.17 ^†^	1.00 (4.00)	1.01
Married/Living with a partner	5.50 (6.00) ^b,c^		1.00 (3.00)	
Divorced	4.00 (3.00) ^c^		1.00 (2.00)	
Nutritional status				
Low weight	4.00 (1.00) ^a,c^	19.26 ^†^	1.00 (1.00)	3.29
Normal weight	7.00 (6.00) ^a^		1.00 (3.00)	
Overweight	9.00 (8.00) ^b,c^		1.50 (4.00)	
Obesity	11.00 (11.50) ^b,c^		2.00 (3.75)	
Occupation				
Studying	8.00 (8.00)	2.28	1.00 (3.00)	0.39
Studying and working	7.00 (8.00)		1.00 (4.00)	
History of COVID-19 infection				
Not yet infected	7.00 (6.75) ^a^	12.70 ^†^	1.00 (3.00) ^a^	6.73 *
Mild symptoms	8.00 (7.00) ^a^		2.00 (4.00) ^a^	
Severe symptoms	14.50 (12.00) ^b^		2.00 (6.75) ^b^	
Family composition				
Living with both parents	8.00 (7.00) ^a,b^	9.24 *	1.00 (3.00) ^a,c,d^	11.96 ^†^
Living with just one parent	8.05 (7.25) ^a^		2.00 (3.50) ^a,b,c^	
Living with other relatives	7.00 (6.50) ^b^		1.00 (2.00) ^a,d^	
Living alone	7.00 (8.00) ^a,b^		2.00 (5.00) ^b^	
Type of university				
Public university	8.00 (8.00)	2.89	1.00 (3.75)	0.81
Private university	7.00 (7.00)		2.00 (4.00)	

* *p* < 0.05, ^†^ *p* < 0.01, ^‡^ *p* < 0.001. ^a–d^ Different superscript letter pairs indicate significant differences between the variable categories. Note: Kruskal–Wallis was used to assess the differences in marital status, history of COVID-19, and family composition. The Bonferroni correction post hoc test was used to assess the significant differences observed in the Kruskal–Wallis. Different subscripts indicate statistically significant differences ^a–d^. The Mann–Whitney test was used to assess the differences in sex, occupation, and university type.

**Table 2 nutrients-16-02901-t002:** Means, standard deviations, and correlation matrix between age, number of months as intern, emotional eating, alcohol consumption, and the various COVID-19’s emotional impact dimensions.

Variables	M (SD)	Correlations								
		1	2	3	4	5	6	7	8	9
1. Age	26.15 (5.60)	-								
2. Number of months as an intern	8.08 (3.31)	0.18 *	-							
3. Emotional eating	8.99 (5.93)	−0.19 *	−0.01	-						
4. Risk of alcohol use disorder	2.65 (3.54)	−0.02	0.06	0.21 *	-					
5. Anger	6.16 (3.53)	−0.14 ^†^	−0.04	0.32 *	0.17 *	-				
6. Fear	5.32 (3.38)	−0.26 *	−0.09 ^‡^	0.37 *	0.07	0.54 *	-			
7. Anxiety	2.72 (2.79)	−0.26 *	−0.04	0.56 *	0.19 *	0.31 *	0.46 *	-		
8. Stress	4.66 (3.47)	−0.26 *	−0.03	0.55 *	0.23 *	0.39 *	0.43 *	0.69 *	-	
9. Depression	3.64 (3.46)	−0.29 *	−0.04	0.59 *	0.23 *	0.31 *	0.41 *	0.66 *	0.80 *	-
10. Overall emotional impact	22.50 (13.00)	−0.30 *	−0.06	0.60 *	0.22 *	0.68 *	0.74 *	0.75 *	0.85 *	0.80 *

* *p* < 0.001, ^†^
*p* < 0.01, ^‡^ *p* < 0.05.

**Table 3 nutrients-16-02901-t003:** Linear regression model estimated to predict emotional eating.

Variables	*B*	*p*
Intercept		<0.001
Age	−0.07	0.092
Sex (Male)	−0.04	0.332
Overall emotional impact	0.24	0.002
Nutritional status (Low weight)	−0.04	0.272
Nutritional status (Overweight)	0.14	<0.001
Nutritional status (Obesity)	0.14	<0.001
Depression	0.24	<0.001
Anxiety	0.17	0.005
Family composition (Living with one parent)	0.10	0.012
Family composition (Living with other relatives)	−0.01	0.689
Family composition (Living alone)	0.01	0.792

*B* = standardized coefficient; *p* = significance. Note: The reference category for sex was female. The reference category for nutritional status was normal weight. The reference category for family composition was living with both parents.

**Table 4 nutrients-16-02901-t004:** Linear regression model estimated to predict alcohol consumption.

Variables	*B*	*p*
Intercept		0.021
Age	−0.09	0.058
Sex (Male)	0.33	<0.001
Depression	0.21	<0.001
Number of months as intern	0.11	0.015
Family composition (Living with one parent)	0.11	0.018
Family composition (Living with other relatives)	−0.01	0.987
Family composition (Living alone)	0.12	0.016
Anger	0.08	0.068
Type of university (Private university)	0.08	0.118
Anxiety	0.09	0.131

*p* = significance; *B* = standardized coefficient. Note: The reference category for sex was female. The reference category for family composition was living with both parents. The reference category for university type was public university.

## Data Availability

The datasets generated or analyzed during this study are available from the corresponding author on reasonable request.

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
