# Peer review of "Emotional Impact of COVID-19 and Emotional Eating and the Risk of Alcohol Use Disorder in Peruvian Healthcare Students"

_nutrients, 2024, doi:10.3390/nu16172901_

Round 1

Reviewer 1 Report

Comments and Suggestions for Authors

The authors conducted a cross-sectional study aimed at exploring the association between the emotional impact of COVID-19, as measured by the COVID-19 Emotional Impact Profile questionnaire, and emotional eating, assessed using the Mindful Eating Questionnaire (MEQ), as well as the risk of alcohol use disorder, evaluated with the Alcohol Use Disorders Identification Test (AUDIT), among Peruvian health science students. The study involved 456 health science interns from four cities in Peru.

I would like to raise the following concerns.

1.

In Table 1, emotional eating and risk of alcohol use disorder are presented as means with standard deviations according to sex, marital status, occupation, history of COVID-19 infection, family composition, and type of university. However, the fact that 3 standard deviations > mean (e.g., emotional eating in females, standard deviation = 5.94 * 3 > mean = 9.29) indicates a non-normal distribution. The authors performed statistical tests using ANOVA and post hoc comparisons with Tukey’s correction, but it might be more appropriate to use the Kruskal-Wallis test with post hoc comparisons instead.

2.

In Table 3, the potential factors include nutritional status, but it does not appear in Tables 1-2.

3.

In Table 3, the linear regression model was used to estimate the prediction of emotional eating. To include the potential factors in the model, it is suggested to provide detailed explanations. Generally, each variable with a significant association (P<0.05) and additional variables that were not significant but had potential clinical importance were introduced into the model.

4.

Similarly, in Table 4, to include the potential factors in the model, it is suggested to provide detailed explanations (e.g., Spearman’s correlations between the number of months as an intern and the risk of alcohol use disorder is 0.06 (p > 0.05, not significant)).

5.

The COVID-19 Emotional Impact Profile seems to be an important factor based on the research objective: 'Emotional impact of COVID-19 and emotional eating and the risk of alcohol use disorder in healthcare students.' However, aside from Table 1, which presents the association between the emotional impact of COVID-19 and emotional eating, as well as the emotional impact of COVID-19 and the risk of alcohol use disorder (an unadjusted statistical analysis), it is not shown in an adjusted model. It is suggested to provide detailed explanations.

6.

It is suggested to reassess the statistical analysis methods used in this study.

Author Response

Dear Reviewer, we sincerely appreciate your comments and suggestions, which have greatly helped us improve our manuscript, especially in the instruments and methods sections. Below, we present our responses to each of your observations:

Comment 1: In Table 1, emotional eating and risk of alcohol use disorder are presented as means with standard deviations according to sex, marital status, occupation, history of COVID-19 infection, family composition, and type of university. However, the fact that 3 standard deviations > mean (e.g., emotional eating in females, standard deviation = 5.94 * 3 > mean = 9.29) indicates a non-normal distribution. The authors performed statistical tests using ANOVA and post hoc comparisons with Tukey’s correction, but it might be more appropriate to use the Kruskal-Wallis test with post hoc comparisons instead.

Response 1: Thank you for your insightful observation. We have reassessed our analytical approach and have now applied non-parametric models, specifically the Kruskal-Wallis test with Bonferroni post hoc comparisons and the Mann-Whitney U test. Additionally, we have updated the statistics reported in Table 1, now presenting medians and interquartile ranges (IQR) for each category. These revisions are thoroughly described in the Analysis Plan section of the manuscript. However, for the regression analyses, we retained the variable transformation approach, as it demonstrated a strong adherence to the assumptions of the general linear model.

Comment 2: In Table 3, the potential factors include nutritional status, but it does not appear in Tables 1-2.

Response 2: We appreciate your observation. Nutritional status has now been included in Table 1, along with the corresponding comparisons. Additionally, we have incorporated a description of the findings related to this factor in the manuscript text. However, we did not include nutritional status in Table 2, as it is a categorical variable that cannot be appropriately analyzed using correlation coefficients with continuous variables.

Comment 3, 4 y 5: In Table 3, the linear regression model was used to estimate the prediction of emotional eating. To include the potential factors in the model, it is suggested to provide detailed explanations. Generally, each variable with a significant association (P<0.05) and additional variables that were not significant but had potential clinical importance were introduced into the model. Similarly, in Table 4, to include the potential factors in the model, it is suggested to provide detailed explanations (e.g., Spearman’s correlations between the number of months as an intern and the risk of alcohol use disorder is 0.06 (p > 0.05, not significant)).The COVID-19 Emotional Impact Profile seems to be an important factor based on the research objective: 'Emotional impact of COVID-19 and emotional eating and the risk of alcohol use disorder in healthcare students.' However, aside from Table 1, which presents the association between the emotional impact of COVID-19 and emotional eating, as well as the emotional impact of COVID-19 and the risk of alcohol use disorder (an unadjusted statistical analysis), it is not shown in an adjusted model. It is suggested to provide detailed explanations.

Response 3, 4 y 5: Thank you for highlighting these important considerations. We have now provided a more comprehensive explanation of the variable selection process, extending beyond the assessment of simple bivariate associations. When incorporating variables into a regression model, the relationships between variables can shift, and as such, relying solely on initial bivariate associations may introduce inaccuracies. To ensure the most robust model, we employed a stepwise selection algorithm, which evaluates the best possible model by considering both statistical significance and the overall fit of the model, rather than relying solely on the significance of bivariate relationships. Additionally, we acknowledge the importance of the COVID-19 Emotional Impact Profile given the study’s objectives and have now included it in the adjusted models as detailed in the revised manuscript.

Comment 6: It is suggested to reassess the statistical analysis methods used in this study.

Response 6: We have thoroughly reassessed the statistical analysis methods used in this study and have expanded our rationale in the Methods section to provide greater clarity and justification for the approaches selected.

We once again thank you for your valuable suggestions and are confident that these improvements will significantly strengthen our work. We remain attentive to any additional comments you may have.

Sincerely,

Reviewer 2 Report

Comments and Suggestions for Authors

Congratulations on your paper.

I have some considerations about your paper:

1.      You should include Peruvian students in the title of your paper

2.      You must include in methods how you calculate emotional eater and explain the measurements

3.      Which questionnaire do you use to measure emotional eating?

4.      From lines 254-265 it is not necessary to include Rho value and signification, because it appears across Table 2

5.      What kind of interventions you propose to reduce emotional impact, could you explain it in the paper?

Author Response

Dear Reviewer, we would like to express our sincere gratitude for the time you have dedicated to reviewing our manuscript and for the valuable insights, recommendations, and suggestions you have provided. These contributions have significantly improved the quality and clarity of our scientific manuscript. Below, we present our responses to each of your observations:

Comment 1: You should include Peruvian students in the title of your paper.

Response 1: We agree with your suggestion and have incorporated the term "Peruvian students" into the title of the paper.

Comment 2 y comment 3: You must include in methods how you calculate emotional eater and explain the measurements. Which questionnaire do you use to measure emotional eating?

Response 2: Thank you for bringing this to our attention. We have now provided a more detailed explanation of the emotional eating scale used in our study. Additionally, we have included information on how we established the categories of emotional eating based on the scale's ranges. These updates can be found in Section 2.3, Measurement Tools, lines 184 to 191.

Comment 4: From lines 254-265 it is not necessary to include Rho value and signification, because it appears across Table 2

Response 4: We agree with your observation and have removed the redundant Rho values from lines 254-265, as suggested.

Comment 5: What kind of interventions you propose to reduce emotional impact, could you explain it in the paper?

Response 5: We agree with your suggestion. We have added, in the discussion section, several interventions that have been shown to be effective in improving psychological well-being among university students. These suggestions can be found in lines 470 to 492.

We once again thank you for your valuable suggestions and are confident that these improvements will significantly strengthen our work. We remain attentive to any additional comments you may have.

Sincerely,

Reviewer 3 Report

Comments and Suggestions for Authors

There are several other aspects that raised my concerns:

- all the measures are self-reported ones, please at least include information about the evaluation of responses collected, strategies used to avoid casual responses, number of people enrolled without responses. 

- the sample has a severe imbalance (83% female) that is a remarkable problem for generalizability. Nothing is said about the sex distribution of the symptoms.

- this is not an explorative study, please consider statistical corrects to avoid error type 1

- how did you evaluate mental health? Why alcoholism was not considered a mental issue (that is an exclusion criterion)?

- the statistical plan reports errors like the use of Student's t-test for categorical levels while it is a tool for continuos variables. Moreover, they used Spearman's correlations because there were non-parametric variables, than authors should use only non-parametric tools.

- is it really possible to use sex in regression models with this sample (where males are just a quarter of the total)?

- data were collected more than 2 years ago, and after 2 years for the beginning of the pandemic. Are these data valuable for anything? This is my major concerns, because the current literature has already shown the presence of specific effects - even alcholism - in several population, but also that this aspect changed after a while. 

Author Response

Dear reviewer. We are grateful to you for your time in helping us to advance whit this opportunity for sharing the findings of our study. We appreciate your comments and suggestions. We have addressed them to the best of our ability as noted below.

Comment 1: all the measures are self-reported ones, please at least include information about the evaluation of responses collected, strategies used to avoid casual responses, number of people enrolled without responses.

Response 1:

We agree with your observation. In response, we have added a paragraph detailing the strategies employed to minimize casual responses. This information can be found in Section 2.1, Sample and Procedures, lines 131 to 137. Additionally, we have included the number of potential participants who were initially invited to take part in the study at the beginning of the Results section.

Comment 2: the sample has a severe imbalance (83% female) that is a remarkable problem for generalizability. Nothing is said about the sex distribution of the symptoms.

Is it really possible to use sex in regression models with this sample (where males are just a quarter of the total)?

Response 2: We acknowledge the point about the sample imbalance, with 83% of the participants being female. This is indeed a significant aspect, particularly in terms of generalizability to male participants. However, I would like to clarify the following:

The sample used in this study is drawn from a population where females are predominant, particularly in healthcare professions such as nursing, medicine, and pharmacy. Thus, while the imbalance is noted, it is reflective of the actual gender distribution within the target population, which justifies its inclusion in the analysis.

Including sex as a variable in the regression model allows us to control for its effect, thereby helping to account for the differences between male and female participants. While the smaller male sample may limit generalizability to some extent, it does not preclude the use of regression analysis. On the contrary, regression models are valuable tools for understanding how predictors, such as sex, influence outcomes, even when there is an imbalance in the sample.

The imbalance in the sample is indeed a limitation that is acknowledged. However, it does not invalidate the findings of the regression analysis. Instead, it suggests that the results are more applicable to the female-dominated population, which is the reality of the healthcare professions being studied. This will be noted in the discussion section to clarify the scope of the findings.

We appreciate your valuable input, which will help strengthen the discussion of the study's limitations and ensure that the interpretation of the results is appropriately contextualized.

Comment 3: this is not an explorative study, please consider statistical corrects to avoid error type 1.

Response 3:

Thank you for your comment. We understand the concern regarding the potential for Type I error in exploratory studies. However, we would like to clarify that the analyses conducted were not exploratory in nature. Initially, we performed descriptive analyses of the correlations, followed by two stepwise regression models focusing on two different dependent variables. In this context, the significance levels were not subjected to multiple comparison adjustments because the primary focus was on the relationships identified within each regression model. As the analyses were hypothesis-driven and specific to our research questions, we determined that Type I error corrections were not necessary. We believe this approach maintains the integrity of the statistical analysis while allowing us to accurately explore the relationships of interest.

Comment 4: how did you evaluate mental health? Why alcoholism was not considered a mental issue (that is an exclusion criterion)?

Response 4:

We partially agree with this comment. In this study, we assessed mental health using the COVID-19 Emotional Impact Profile questionnaire, which measures individuals’ levels of anger, resentment, fear, anxiety, stress, and depression. To address your suggestion, we have improved the wording in this section for greater clarity. Regarding the exclusion of alcoholism as a mental health issue, while we recognize that alcoholism could be considered a mental health problem, we chose not to include it as such because our focus was on measuring the risk of alcohol use rather than diagnosing alcoholism. Additionally, our primary interest was to explore how the most common psycho-emotional issues reported during the COVID-19 pandemic influenced eating behaviors and the likelihood of adopting harmful habits, such as alcohol consumption.

Comment 5: The statistical plan reports errors like the use of Student's t-test for categorical levels while it is a tool for continuos variables. Moreover, they used Spearman's correlations because there were non-parametric variables, than authors should use only non-parametric tools.

Response 5: Thank you for your careful review of our statistical plan. We acknowledge the importance of using appropriate statistical tests and have taken steps to correct any inaccuracies. Specifically, we have reassessed our analytical approach and have replaced the use of the Student's t-test with non-parametric models where appropriate. We now apply the Kruskal-Wallis test with Bonferroni post hoc comparisons and the Mann-Whitney U test for the analysis of categorical variables, as these are better suited for non-parametric data. Additionally, we have updated the relevant statistics in the manuscript, presenting medians and interquartile ranges (IQR) instead of means and standard deviations where applicable. These changes are detailed in the revised Analysis Plan section of the manuscript. However, for the regression analyses, we retained the variable transformation approach, as it met the assumptions required for the general linear model.

Comment 6: data were collected more than 2 years ago, and after 2 years for the beginning of the pandemic. Are these data valuable for anything? This is my major concerns, because the current literature has already shown the presence of specific effects - even alcholism - in several population, but also that this aspect changed after a while.

Response 6:

Thank you for this pertinent comment, which prompted us to reflect as a team on the implications of our study. As a result of this discussion, we have added a paragraph in the Discussion section, lines 473 to 498, which we hope addresses your major concerns. While the COVID-19 pandemic has ended, other conditions such as depression, anxiety, nutritional status, and the duration of internships continue to impact the mental health of students. Therefore, it remains essential to highlight these issues so that universities can implement preventive interventions.

We once again thank you for your valuable suggestions and are confident that these improvements will significantly strengthen our work. We remain attentive to any additional comments you may have.

Sincerely,

Round 2

Reviewer 1 Report

Comments and Suggestions for Authors

No further comment  

Author Response

Dear Reviewer

We are grateful to you for your time in helping us to advance with this opportunity for sharing the findings of our study.

We thank you for your continued interest in our research.

Sincerely,

Jhon Zeladita

Reviewer 3 Report

Comments and Suggestions for Authors

The authors have partially addressed my concerns about the manuscript. While they have corrected the statistical issues, the methodological problems remain. They didn't clarify how they evaluated mental health (despite this being an exclusion criterion) and no specific strategy to avoid automatic responders had been used. 

However, there are still some problems with the statistics. Table 1 still reported "Student’s t-tests were used to assess the differences in sex, occupation, and university type" and they are all categorical variables, how is possible to use Student's t? 

My concerns about the manuscript's coherence with the journal's aims remain. I think the paper is out of the scope of the journal and also reported data that presents serious flows and problems for generalization. 

Author Response

Reviewer:

Dear Reviewer, we sincerely appreciate your comments on our manuscript. Below, we present our responses to each of your observations:

Comment 1: The authors have partially addressed my concerns about the manuscript. While they have corrected the statistical issues, the methodological problems remain. They didn't clarify how they evaluated mental health (despite this being an exclusion criterion) and no specific strategy to avoid automatic responders had been used.

Response 1: We appreciate the reviewer’s clarification regarding the exclusion criterion for mental health. It should be noted that, concerning this criterion, students were asked whether they had ever been diagnosed with a mental health condition by a doctor.

Regarding the strategy to avoid automatic responses, we developed an informed consent form that explained the importance of honest participation in the study. Additionally, we ensured that responses were collected only from health students currently in internships. It is also important to consider that during the study period, multiple online studies were conducted, leading to more conscious participation from both students and researchers.

Although it is true that, despite the precautions mentioned above, we cannot affirm with complete certainty that automatic responses were entirely avoided, it is important to note that such responses could introduce errors, which might be reflected in inconsistent or unexpected results. However, the alignment of our findings with existing scientific literature and established logic suggests that if there were automatic responses that we were unable to detect, they likely did not have a significant impact on the estimated models. We consider this a limitation of our research, although it does not significantly affect the validity of our findings.

Comment 2: However, there are still some problems with the statistics. Table 1 still reported "Student’s t-tests were used to assess the differences in sex, occupation, and university type" and they are all categorical variables, how is possible to use Student's t?

Response 2: We have reviewed this observation and respectfully disagree with the reviewer. In Table 1, we present the comparison of the scores (medians) for the emotional eating and alcohol consumption tests according to population characteristics such as gender, occupation, and type of university. For dichotomous variables, we applied the Mann-Whitney U test, and for polytomous variables, we used the Kruskal-Wallis test to assess differences in scores based on these characteristics. This procedure is recommended when variables do not follow a normal distribution.

We appreciate the reviewer's comment, which helped us identify that, by mistake, the Student's T-test was still listed at the bottom of Table 1 in the previously submitted version. However, given the non-normal distribution of the data, the Mann-Whitney U test was the appropriate method applied.

The title of Table 1 has been modified to: “Comparison of Medians of Emotional Eating and alcohol consumption according to sex, Marital Status, occupation, history of COVID-19 Infection, Family Composition, and Type of University”.

Comment 3: My concerns about the manuscript's coherence with the journal's aims remain. I think the paper is out of the scope of the journal and also reported data that presents serious flows and problems for generalization.

Response 3: In this comment we also respectfully disagree with the reviewer. We consider that the article provides relevant information on dietary and alcohol consumption behaviors as an impact or consequence of the pandemic, given that these topics have been explored in other areas and study groups published in the journal. We hope that this clarification addresses the reviewer's concerns. It is important to note that our findings are limited to the study population and are not intended to generalize.

We once again thank you for your valuable suggestions. We remain attentive to any additional comments you may have.

Sincerely,
